# Prognostic Role of the Intrahepatic Lymphatic System in Liver Cancer

**DOI:** 10.3390/cancers15072142

**Published:** 2023-04-04

**Authors:** Katsunori Sakamoto, Kohei Ogawa, Kei Tamura, Masahiko Honjo, Naotake Funamizu, Yasutsugu Takada

**Affiliations:** Department of Hepato-Biliary-Pancreatic and Breast Surgery, Ehime University Graduate School of Medicine, 454 Kou, Shitsukawa, Toon 791-0295, Ehime, Japan

**Keywords:** live cancer, lymphatic vessel invasion, lymphangiogenesis, lymph node metastasis, hepatocellular carcinoma, intrahepatic cholangiocarcinoma, colorectal liver metastasis

## Abstract

**Simple Summary:**

The prognostic impact of intrahepatic lymphatic vessel invasion (LVI) in liver cancer has rarely been reported. We sought to clarify the prognostic impact of intrahepatic lymphatic system involvement in liver cancer. Tumor-associated lymphangiogenesis reportedly correlates with prognosis after HCC resection. A meta-analysis showed that overall survival was poorer in patients with positive LVI than with negative LVI after resection of ICC and colorectal liver metastasis. Lymphangiogenesis was also reported to predict unfavorable prognosis in ICC. A few reports showed correlations between LVI/lymphangiogenesis and LNM in liver cancer. LVI and lymphangiogenesis showed worse prognostic impacts for liver cancer than their absence, but further study is needed.

**Abstract:**

Although several prognosticators, such as lymph node metastasis (LNM), were reported for hepatocellular carcinoma (HCC) and intrahepatic cholangiocarcinoma (ICC), the prognostic impact of intrahepatic lymphatic vessel invasion (LVI) in liver cancer has rarely been reported. We sought to clarify the prognostic impact of intrahepatic lymphatic system involvement in liver cancer. We systematically reviewed retrospective studies that described LVI and clinical outcomes of liver cancer and also included studies that investigated tumor-associated lymphangiogenesis. We conducted a meta-analysis using RevMan software (version 5.4.1; Cochrane Collaboration, Oxford, UK). The prognostic impact of intrahepatic LVI in HCC was not reported previously. However, tumor-associated lymphangiogenesis reportedly correlates with prognosis after HCC resection. The prognostic impact of intrahepatic LVI was reported severally for ICC and a meta-analysis showed that overall survival was poorer in patients with positive LVI than with negative LVI after resection of ICC. Lymphangiogenesis was also reported to predict unfavorable prognosis in ICC. Regarding colorectal liver metastases, LVI was identified as a poor prognosticator in a meta-analysis. A few reports showed correlations between LVI/lymphangiogenesis and LNM in liver cancer. LVI and lymphangiogenesis showed worse prognostic impacts for liver cancer than their absence, but further study is needed.

## 1. Introduction

The liver produces 25–50% of total lymph fluid in the body and includes a dense network of lymphatic vessels linking to the cisterna chyli and thoracic duct [1,2,3]. The lymphatic system plays critical roles in fluid homeostasis and immune response [1,2,3]. Lymphatic vessels related to the liver can be classified into the following three categories: portal, sublobular, and capsular (superficial) [1,2,3]. Portal lymphatic vessels run within portal tracts surrounded by a Glissonean sheath and extend to the terminal portal tract [1,2,3,4]. Portal lymphatic vessels also connect with other portal tracts via short side branches [1,2,3]. Sublobular lymphatic vessels distribute throughout the hepatic sinusoids, space of Disse (perisinusoidal space), and connecting channels between the space of Disse and perihepatic interstitial tissue [1,2,3]. Sublobular lymphatic vessels lead into portal lymphatic vessels or lymphatic vessels running alongside the central veins [1,2,3]. Sinusoids are specific to the liver and consist of a single layer of liver sinusoidal endothelial cells without any basement membrane [2]. Hepatic lymph fluid originates from plasma components that filter through the fenestrae of sinusoids into the space of Disse [2]. As mentioned above, fluid in the space of Disse flows into portal lymphatic vessels or lymphatic vessels running alongside the hepatic veins [2]. Capsular lymphatic vessels distribute directly underneath the liver capsule, as suggested by the name [1,2,3]. Approximately 80% or more of hepatic lymph drains through portal lymphatic vessels to the lymph nodes of the hepatic hilum and lesser omentum; then, these lymph nodes connect to celiac nodes to drain into the cisterna chyli and thoracic duct [1]. The remaining lymph (20% or less) drains through the sublobular and capsular lymphatic vessels [1]. Capsular lymphatic vessels of the convex surface of the liver drain into mediastinal lymph nodes through the coronary ligament, whereas those of the concave surface drain into the lymph nodes of the hepatic hilum and regional lymph nodes [2]. Portal lymph flow increases in situations of high portal vein pressure, such as liver cirrhosis, contrasting with the decrease in portal blood flow.

Liver cancer is the fourth leading cause of cancer-related deaths worldwide [3]. In 2015, approximately 854,000 cases of liver cancer were diagnosed and 810,000 patients died [5]. The most common type of primary liver cancer is hepatocellular carcinoma (HCC), followed by intrahepatic cholangiocarcinoma (ICC), comprising approximately 80% and 15%, respectively [5,6,7]. For HCC without cirrhosis, surgical resection offers the best chance of cure, but most cases are unfortunately diagnosed at an advanced stage that reduces the chances of successful surgery [7]. Furthermore, the recurrence rate remains high even if a patient with HCC undergoes curative resection, with 5-year recurrence rates of up to 70% [8,9,10]. Similarly, the prognosis of ICC is also very poor even after curative resection, with 5-year overall survival (OS) rates of 15–40% [11,12]. Many prognostic factors were identified for HCC and ICC; of those, vascular invasion to the portal and hepatic veins is widely recognized as a strong prognosticator for HCC and ICC [13,14,15,16,17,18,19,20,21,22,23,24,25,26,27,28,29,30,31,32]. However, the prognostic impact of intrahepatic lymphatic vessel invasion (LVI) was rarely reported for HCC and was only described in a few reports for ICC [33]. Regarding secondary liver cancers, few reports evaluated the prognostic impact of intrahepatic LVI for colorectal live metastasis (CRLM) [34,35,36,37]. When not limited to liver cancer, LVI was reported as a major prognostic marker for pathologies such as breast cancer, endometrial cancer and colon cancer, with significant correlations to lymph node metastasis (LNM) [38,39,40,41,42,43]. However, although the liver produces and contains a large amount of lymph fluid, studies evaluating the prognostic roles of the intrahepatic lymphatic systems in liver cancer remain insufficient.

One of the major concerns regarding intrahepatic LVI in liver cancer is the increasing incidence of LNM. The prognostic impacts of LNM such as hepatic nodes in HCC and ICC were widely reported, and a positive finding is considered a strong poor prognosticator [11,44,45,46,47,48,49,50,51]. However, the association between intrahepatic LVI and LNM and the mechanisms involved were not sufficiently investigated [52]. Regarding the prognostic role of lymphatic system, not only LVI but also peritumoral lymphangiogenesis is reported as prognostic factor of [53]. Herein, we, therefore, systematically reviewed studies that investigated the prognostic impact of intrahepatic lymphatic systems in liver cancer, with a view to promote further studies of the associations between intrahepatic lymphatic systems and liver cancers.

## 2. Methods

### Literature Search and Meta-Analyses

We systematically reviewed studies that provided information on both LVI and clinical outcomes in accordance with the Preferred Reporting Items for Systematic Reviews and Meta-Analyses (PRISMA) guidelines [54]. We also included studies that investigated tumor-associated lymphangiogenesis, which is the sprouting of new lymphatic vessels in the tumor microenvironment, as a form of tumor-associated neovascularization that was considered to represent the metastatic spread of highly aggressive forms of cancer [55,56]. Literature searches of the Cochran database and PubMed were performed for articles published from January 2000 to February 2023. The search strategy “((liver tumor) OR (liver cancer) OR (hepatocellular carcinoma) OR HCC OR (intrahepatic cholangiocarcinoma) OR ICC OR IHCC OR (liver metastasis) OR (metastatic liver tumor)) AND ((Lymphatic vessel invasion) OR (lymph vessel invasion) OR lymphangiogenesis OR (lymph vessel density) OR (lymphatic vessel density) OR podoplanin OR D2-40)” was used and developed by one reviewer (KS). Study selection was performed by two reviewers (KS, KT). Additionally, a manual search of the references of included studies in this review was conducted and the studies which were suitable for this review were also included. Because the studies investigating the prognostic impact of LNM in liver tumors are enough reported, we excluded the studies that only focused on LNM and included the studies that combined an assessment of LVI/lymphangiogenesis. We also excluded the studies about hilar/perihilar cholangiocarcinoma, as most of the main lesions are in extrahepatic bile duct. All studies we identified were retrospective in design.

We conducted a meta-analysis using RevMan software (version 5.4.1; Cochrane Collaboration, Oxford, UK). Dichotomous outcomes are shown as risk differences and 95% confidence intervals. Heterogeneity among the included trials was evaluated using a forest plot. I-squared and chi-squared statistics were used to evaluate statistical heterogeneity [57,58]. A random-effects model was used [59].

## 3. Results

### 3.1. Intrahepatic Lymphatic Vessels

Intrahepatic lymphatic vessels such as sublobular or portal vessels are difficult to detect macroscopically during surgery. However, capsular lymphatic vessels can be detected macroscopically using laparoscope-enhanced views (Figure 1). Regarding microscopic findings, although the lymphatic vessels and portal veins can be difficult to distinguish from each other using standard histochemical techniques, the anti-podoplanin antibody D2-40 is specific for lymphatic vessel endothelium and enables the detection of intrahepatic lymphatic vessels [60]. Regarding lymphangiogenesis, lymphatic vessel counts from microscopic findings were used to determine lymphatic vessel density (LVD) [53,61,62]. In most studies describing methods for the assessment and quantification of LVD, the lymphatic vessel numbers counted manually at 200× magnification (a 0.25-mm^2^ field) in several areas of highest vascular density identified in low-magnification (×25–40) views were used for assessing LVD [60,61]. We found that D2-40 was consistently used for detecting lymphatic vessels in studies published since 2007.

### 3.2. Intrahepatic Lymphatic Vessels and Liver Cirrhosis

Yokomori et al. also reported that not only capsular lymphatic vessels, but also non-capsular lymphatic vessels, are enlarged in cirrhotic patients due to the high portal pressure [60]. The high portal pressure with increased sinusoidal blood flow is caused by architectural deformations around the portal and central veins [3]. Lymph fluid production in the liver is increased up to 30-fold in cirrhotic patients [3]. This represents the main cause of ascites, which comprises fluid that leaked from lymphatic vessels [2,56]. Furthermore, the number of lymphatic vessels also increased in cirrhotic liver [4]. These changes are due to the expression of lymphangiogenic growth factors such as vascular endothelial growth factor (VEGF)-C or VEGF-D due to the processes of tissue repair, inflammation, and tumor-related factors [4].

### 3.3. HCC

The long-term prognostic impact of intrahepatic LVI in HCC was not investigated previously. The incidence and prognostic impact of intrahepatic LVI in HCC are thus unclear. Two patients who underwent liver transplantation and showed lymphangiosis carcinomatosa in the resected specimens showed relatively good prognosis, with both surviving without recurrence (39 months and 16 months) after transplant [63]. In contrast, tumor-associated lymphangiogenesis reportedly correlated with prognosis after the resection of HCC [53]. Thelen et al. quantified peritumoral intrahepatic LVD using D2-40 for patients who underwent hepatectomy for HCC, revealing that tumors with high LVD were associated with poor OS (Table 1) [53]. High LVD was defined as >22.9 vessels in a 200× view (0.25-mm^2^ field) [53]. The high-LVD group showed a poorer disease-free survival (DFS) rate than the low-LVD group (18% vs. 40%, respectively; *p* = 0.047) and LVD was selected as an independent predictor of DFS (but not of OS) in multivariate analysis [53]. Lymphatic vessels were detected in the liver stroma in both healthy and cirrhotic livers, whereas HCC exhibited lymphatic vessels in both the tumor parenchyma and intratumoral septa [53]. Cioca et al. reported that high expression of podoplanin in tumor cells was associated with higher frequency of poorly differentiated histopathological type than that of low in HCC (59% vs. 41%, respectively; *p* = 0.040), suggesting a role of podoplanin in hepatocarcinogenesis [64]. Furthermore, high peritumoral LVD correlated with both cirrhosis and vascular invasion (*p* = 0.006 and *p* = 0.018, respectively) [64]. In another study, lymphangiogenesis-related long non-coding RNAs were able to be used to estimate HCC prognosis (median survival time [MST]: approximately 2 years for long non-coding RNA pairs with high-risk group vs. 7 years for those with low-risk group, *p* < 0.001) and may be useful to select candidates for anti-tumor immunotherapy and chemotherapy [65].

Several molecular and genetic mechanisms, such as HCC-associated long noncoding RNA (HANR), VEGF-C and -D, VEGF receptor (VEGFR)-3, heparanase-1, lymphatic vessel endothelial hyaluronan receptor-1 (LYVE-1), hypoxia-inducible factor (HIF)-2a and homeobox prospero-like protein-1 (Prox-1), showed correlations with peritumoral lymphangiogenesis in HCC and associations with a greater risk of metastasis [1,66,67,68,69,70,71].

### 3.4. ICC

The prognostic impact of intrahepatic LVI was reported in much greater detail for ICC than for HCC (Table 2) [38,61,72,73,74]. Nakajima et al. reported on 16 patients with LVI among 102 ICC patients in 1988 [75]. While they identified LNM in 12 (75%) of the 16 LVI-positive patients [67], the incidence of LNM was similar to that of LVI-negative patients (70%) and no significant correlation between LVI and LNM was detected [75]. Fisher et al. reported not only LVI, but also perineural invasion as independent prognosticators of OS [38]. Lang et al. reported that although LVI was associated with significantly poorer OS in univariate analyses, R0 resection and cancer stage, but not LVI, remained as independent predictors of OS in multivariate analyses [74]. Conversely, Cho et al. reported significant differences in MST, at 9 months in LVI-positive patients and 23 months in LVI-negative patients (*p* = 0.008) and selected LVI as an independent predictor of OS [73]. Recently, Lurje et al. reported that LVI-positive patients after ICC resection showed shorter MST than LVI-negative patients in a univariate analysis (Table 2) [72]. However, although multivariate analysis in their study showed LVI as an independent predictor of OS in the perihilar cholangiocarcinoma cohort, LVI was not selected as an independent predictor of OS in the ICC cohort [72]. A meta-analysis showed that positive LVI in patients with resected ICC was associated with poorer OS than negative LVI (Figure 2). Five-year OS was 35.3% in LVI (−) group and 13.4% in LVI (+) group (risk difference −0.22; 95% confidence interval −0.40 to −0.04; *p* = 0.01, Figure 2). Shirabe et al. and Patel et al. also reported the poor prognostic impact of LVI on ICC, but those studies were excluded from our meta-analyses because the patient cohorts might have represented duplicates of those in studies by Aishima et al. and Fisher et al., respectively [33,76]. Furthermore, another report that included 26 patients with ICC and 14 patients with perihilar cholangiocarcinoma who underwent liver transplant found that pathological LVI was associated with poorer recurrence-free survival (hazard ratio, 2.1) [77]. In contrast, Yoshikawa et al. reported that LVI had no significant impact on prognosis in ICC [78]. They reported that epidermal growth factor receptor (EGFR) was associated with poor prognosis and represented an independent prognosticator in multivariate analyses (5-year OS 17.7% in EGFR-positive patients vs. 47.1% in EGFR-negative patients, *p* = 0.0008) [78].

**Table 1 cancers-15-02142-t001:** Studies evaluating lymphangiogenesis in liver cancer.

	Author	Published Year	Study Period	LVD	Patient Number	Tumor Size, cm	Tumor Number	Vascular Invasion	Poor Differential Pathology	Cirrhosis	LNM	5-year OS	*p*
HCC	Thelen et al. [53]	2009	1997–1998	>22.9 vessels (200× view)	24	T1/2, 5 (21%)	multiple, 8 (33%)	11 (46%)	5 (21%)	16 (67%)	2 (3%)	24%	0.018
				≤22.9 vessels (200× view)	36	T1/2, 12 (33%)	multiple, 11 (31%)	18 (50%)	7 (19%)	8 (22%)	NR	56%	
ICC	Thelen at al. [62]	2010	NR	>12.66 vessels (200× view)	46	>5, 32 (70%)	multiple, 21 (46%)	13 (28%)	17 (37%)	NR	27 (59%)	6.5%	<0.001
				≤12.66 vessels (200× view)	68	>5, 49 (72%)	multiple, 21 (31%)	17 (25%)	27 (40%)		26 (38%)	31%	
	Sha et al. [79]	2019	2007–2015	≥13 vessels (400× view)	50	≥5, 32 (64.0%)	multiple, 11 (22.0%)	13 (26.0%)	27 (54%)	24 (40%)	32 (64.0%)	0%	<0.001
				<13 vessels (400× view)	56	≥5, 29 (51.8%)	multiple, 6 (10.7%)	15 (26.8%)	27 (48.2%)	NR	16 (28.6%)	48%	

HCC, hepatocellular carcinoma; ICC, intrahepatic cholangiocarcinoma; LNM, lymph node metastasis; LVD, lymphatic vessel density; NR, not reported; OS, overall survival.

Lymphangiogenesis was reported as a possible prognostic marker for HCC, and also as a predictor of unfavorable prognosis for ICC (Table 1) [62,79]. Both studies also reported that high LVD was associated with the presence of LNM and poorer OS [62,79]. The cut-off value between high and low LVD was set at a count of 13 vessels in a 400× view for the study of Sha et al. [79] and 12.66 vessels in a 200× view for that of Thelen et al. [62]. Although the report by Sha et al. [79] concluded that high LVD represented an independent predictor of OS in multivariate analyses, the report by Thelen et al. [62] selected vascular invasion, not high LVD, as an independent predictor of OS. A different study by Sha et al. [80] reported that expression of VEGFR-3, which promotes lymphangiogenesis, correlated with dismal prognosis in ICC. The 5-year OS was 14.6% in VEGFR-3-positive patients, compared to 53.2% in VEGFR-3-negative patients (*p* < 0.001). Aishima et al. also reported that 5-year OS was significantly poorer in VEGF-C-positive patients (0%) than in VEGF-C-negative patients (49%, *p* = 0.0007) [61]. Several molecular mechanisms promoting lymphangiogenesis were reported, such as thrombospondin 1 and 2, C-X-C motif chemokine receptor 2 (CXCR2)-CXC ligand 5 (CXCL5) signaling and platelet-derived growth factor-D [81,82,83,84]. Liu et al. reported that the herbal medicine oxyresveratrol can prevent LNM by inhibiting lymphangiogenesis [85]. Intratumoral lymphangiogenesis was correlated with LNM and prognosis in hilar cholangiocarcinoma [82]. They reported that the frequency of a LNM-positive state was 68% in high-LVD patients and 12% in low-LVD patients (*p* < 0.001), while 5-year OS rates were 7.0% and 76.4%, respectively (*p* < 0.001) [86].

### 3.5. CRLM

In terms of CRLM, several studies investigated the intrahepatic lymphatic system and clinical outcomes (Table 2) [34,35,36,37]. Because of the rarity of candidates for resection of liver metastases from other origins, the present study did not identify any other reports evaluating LVI as a prognostic factor besides those involving CRLM. Among cases with CRLM, LVI represented a poor prognosticator in the meta-analysis (Figure 3). The meta-analysis showed significantly better 5-year OS in LVI (−) group than that of LVI (+) (52.5% vs. 21.3%; risk difference −0.39; 95% confidence interval −0.50 to −0.28; *p* < 0.00001 Figure 3). Sasaki et al. and Korita et al. reported dismal prognosis among LVI-positive patients, with 5-year OS rates of 0% [35,37]. Korita et al. also reported that LVI was significantly associated with LNM [37]. While Lupinacci et al. reported that vascular invasions other than LVI were unrelated to recurrence or survival [36], de Ridder et al. reported that LVI in combination with other vascular invasions is an important sign of adverse prognosis [34].

Schoppmann et al. reported significant correlations between LVI and lymphangiogenesis at both primary colorectal sites and liver metastatic sites [87]. LVD was greater in LVI-positive patients than in LVI-negative patients (*p* = 0.0001) [87]. They concluded that the lymphatic pathway represents a key route for the metastasis of colorectal cancer to the liver, along with the hematogenous pathway [87]. They also reported that high tumor expression of VEGF-C, which promotes lymphangiogenesis and metastasis [41,87,88,89], was associated with poor prognosis in patients with CRLM (MST: approximately 1 year in VEGF-C-positive patients vs. 2 years in VEGF-C-negative patients, *p* = 0.010) [87]. Various lymphangiogenic gene expressions and molecular mechanisms (VEGF-C, neuropilin-2 [Nrp-2], podoplanin, LYVE-1, mannose receptor-C type 1 (MRC1), chemokine (C-C) ligand 21 [CCL-21]) were associated with poor prognosis in CRLM [90]. High expression of VEGF-C and Nrp-2 was also associated with lymph node recurrence following CRLM resection [90].

### 3.6. Association between LVI and LNM

Two main routes of cancer metastasis were reported: hematogenous spread and lymphogenous spread [56]. The route most likely to result in metastasis can depend on tumor factors, the peritumoral microenvironment and patient status [56]. With hematogenous spread, tumor cells directly enter blood vessels and disseminate to distant sites. With lymphogenous routes, tumor cells penetrate into lymphatic vessels and disseminate to regional or distant lymph nodes. Tumor cells in lymph nodes then enter the thoracic duct and move from there to the subclavian vein to metastasize to distant sites [56]. Although the mechanisms from LVI to LNM are known, little work was carried out on correlations between LVI and LNM in liver cancer. Korita et al. revealed an association between LVI and hepatic nodes in CRLM, with a higher incidence of hepatic node involvement in LVI-positive patients (23%) than in LVI-negative patients (4%, *p* = 0.039) [37]. They suggested that LVI in patients with liver metastasis spread to regional lymph nodes via the hepatic networks of portal, sublobular and capsular lymphatic vessels [37]. They termed this phenomenon “remetastasis”, since liver metastases are mainly spread hematogenously, but liver metastases with LVI may spread to LNs via lymphogenous routes [37].

High-level lymphangiogenesis was also associated with a significantly higher incidence of LNM than low-level lymphangiogenesis in ICC in studies by Sha et al. and Thelen et al. (Table 1: 64.0% vs. 28.6%, *p* < 0.001 and 59% vs. 38%, *p* < 0.001, respectively) [79]. Tumor cells can easily migrate along tumor-associated lymphatic vessels into lymph nodes [79]. As mentioned earlier, lymphangiogenesis also correlates with LNM in hilar cholangiocarcinoma [86]. Nevertheless, LNM is widely known as a strong prognosticator of OS in both HCC and ICC [44]. In pooled analyses of HCC from a systematic review by Amini et al., 3- and 5-year OS rates were 27.5% and 20.8% in LNM-positive patients and 60.2% and 42.6% in LNM-negative patients [44]. With ICC, 3- and 5-year OS rates were 0.2% and 0% in LNM-positive patients and 55.6% and 45.1% in LNM-negative patients [44].

## 4. Discussion

A large-scale study of the intrahepatic lymphatic systems in terms of liver cancer prognosis was rarely reported. However, the invasion of lymphatics may have a clinical prognostic impact as strong as that of blood vessel invasion in liver cancer, since the liver manages almost half the lymph generated in the human body [1,2,3,4]. Figure 1 shows the capsular lymphatics, suggesting that tumors invading these vessels may readily metastasize to other organs. The present study, therefore, reviewed the prognostic impact of LVI and lymphangiogenesis in liver cancer, confirming that patients with LVI or prominent lymphangiogenesis showed poorer prognosis than those without such findings.

With regard to HCC, no studies appeared to have evaluated LVI. This may be due to the scarcity of LVI itself in HCC as compared to ICC, because of the reduced invasiveness of HCC compared to ICC [13]. However, in contrast to LVI, one study showed that tumor-associated lymphangiogenesis was associated with poorer prognosis in HCC patients [53]. The same investigation found that lymphangiogenesis was more frequent in cirrhotic patients (66%) than in non-cirrhotic patients (22%, *p* = 0.0001) [53]. Yokomori et al. also showed that lymphangiogenesis was promoted in cirrhotic patients [60]. Since the lymphatic system is associated with immune function, dysfunction of the system caused by cirrhosis or tumor will decrease the ability of the immune system to deal with cancer [82]. The association between HCC and the intrahepatic lymphatic system might, thus, be more important in cirrhotic patients than non-cirrhotic patients.

Several reports found that microscopic serosal invasion in HCC resulted in poor prognosis [13,91,92]. Those reports discussed the possibility that serosal invasion developed to LVI and subsequent distant metastasis (Figure 1) [13,91,93]. Actually, patients with serosal invasion showed poor prognosis not only in terms of direct invasion to other organs, but also in microscopic serosal invasion [13,91,93]. However, those reports investigating serosal invasion in HCC did not perform evaluations of LVI or lymphangiogenesis using D2-40, and so, this may represent an important avenue for future research. Actually, in contrast to other malignancies such as colorectal cancer, gastric cancer, extrahepatic biliary-tract cancer, and breast cancer [38,39,40,41,42,43,86,93,94,95], the pathological evaluations of LVI and lymphangiogenesis are not defined in the liver cancer classification such as the Japanese General Rules for the Clinical and Pathological Study of Primary Liver Cancer [96]. LVI and lymphangiogenesis might thus warrant clear definitions for pathological evaluation, matching those for other gastroenterological cancers.

Although LVI is easy to consider as the main cause of LNM, the correlations between LVI and LNM were not clarified in HCC. This is largely attributable to the low incidence of LNM in HCC, at 0–7.45% in operable cases [97,98,99,100,101,102,103,104]. Hasegawa et al. observed a pathological LNM-positive status in only 112 patients (0.8%) among a large cohort of 14,872 patients with resected HCC [50]. In contrast, LNM rates as high as 30.3% were observed in autopsy cases from Japan [98]. Nevertheless, the prognosis of LNM in HCC patients appeared poor, with 3-year OS rates of 13.6–38.9% [45,46,51,97]. However, the prognostic impact of lymph node dissection remains unclear [45]. Since routine lymph node dissection is not warranted in hepatectomy for HCC due to the low incidence of LNM [47], further studies to identify appropriate candidates for lymph node dissection are required.

Regarding ICC, several reports evaluated LVI in terms of prognostic impact (Table 2, Figure 2) [33,38,61,72,73,74,75,76]. Although the studies included in meta-analysis showed a little heterogeneity (I^2^ = 66%), LVI (+) showed significantly poorer 5-year OS than that of LVI (−) (*p* = 0.01, Figure 2). Therefore, we should focus more on intrahepatic pathological LVI in ICC patients. However, as with HCC, the association between LVI and LNM was not reported. In contrast, although reports remain few in number, lymphangiogenesis was correlated with LNM in ICC and perihilar cholangiocarcinoma [79,86]. The incidence of LNM in the studies selected for this review paper was high, ranging from 22% to 40% [38,61,72,73,74,75], similar to recent large-scale data for LNM in ICC [51,105]. In those two large studies, 391 (34.1%) of 1147 patients who underwent lymphadenectomy had pathologically confirmed metastasis [105] and 249 (41.3%) of 603 patients showed LNMs [51]. As a result, the correlation between LVI and LNM may warrant further investigation in the future. Although routine regional lymphadenectomy for ICC is recommended in the clinical practice guidelines of the National Comprehensive Cancer Network (NCCN), European Society for Medical Oncology (ESMO) and European Association for the Study of the Liver (EASL) [11,49,106], the contribution of this measure to improving long-term prognosis is unclear and consensus remains elusive [47,47,49,105,107,108]. Further study is, therefore, needed to determine appropriate selection criteria for lymphadenectomy, including the prediction and diagnosis of LVI and lymphangiogenesis using intraoperative frozen biopsy, serum markers or gene status [109,110,111,112]. Many genetic and molecular factors associated with lymphangiogenesis were reported for liver cancer [1,66,67,68,69,70,71,81,82,83,84,90]. Positivity for such factors might prove suitable as candidate selection criteria for simultaneous liver resection and lymph node dissection in liver cancer patients, allowing more tailored medicine in the future.

Regarding CRLM, two routes of metastasis are known: the lymphatic route, such as secondary to initial colonization of regional lymph nodes, and direct hematogenous spread from the primary tumor, which is independent of lymph node metastatic growth [113,114]. First, the key evidence for the lymphatic route of metastasis is the high rate of liver metastasis coinciding with LNM [115]. Second, lymphatic vessel density in the primary tumor correlates with both LNM and liver metastasis [116]. Third, since lymphatic vessels have a discontinuous basement membrane and lack coverage by pericytes [117], tumor cells can more easily invade lymphatic vessels. The evidence in support of the other, hematogenous route of metastasis to the liver is: first, many patients with CRLM do not show LNM [115] and second, venous invasion of the primary tumor is an independent prognosticator of CRLM [118]. The prognosis of patients who underwent hepatectomy for CRLM was poorer in patients with LNM of either regional nodes around the primary lesion or hepatic nodes than in those without such LNM [119,120,121,122]. As a result, treatment strategies that consider the specific route to liver metastasis using LVI or lymphangiogenesis may contribute to better selection criteria for hepatectomy in patients with CRLM. Actually, the present meta-analysis without heterogeneity (I^2^ = 43%) showed significantly poor 5-year OS in LVI (+) patients than that of LVI (−) in CRLM patients (*p* < 0.00001, Figure 3). Therefore, further study evaluating LVI/lymphangiogenesis and prognosis of CRLM may develop the tailored surgical procedure such as appropriate hepatic lymph node dissection or precise hepatectomy in the future.

This study showed several limitations. First, the number of patients was insufficient. Second, the accuracy of LVI determinations may have been lacking, as not all the included studies explained the details of the pathological diagnosis for LVI or lymphangiogenesis, such as the use of D2-40. Third, details of LVI, such as the types of lymphatic vessels involved, were also not described in the majority of papers. More precise evaluations of LVI and lymphangiogenesis are warranted in future research to develop more appropriate treatment strategies, including perioperative chemotherapy or lymphadenectomy.

## 5. Conclusions

LVI and lymphangiogenesis have prognostic impacts for liver cancer, but the correlations with LNM remain unclear. Further large-scale studies are warranted. In other words, to reveal the association between LVI/lymphangiogenesis and LNM/prognosis, thorough pathological evaluation of LVI/lymphangiogenesis using D2-40 antibody should be performed in liver cancer as well as other gastroenterological cancer. However, the definition of LVI/lymphangiogensis needs further discussion. Nevertheless, if the prognostic role of intrahepatic lymphatic system and mechanism of lymphatic spread of carcinoma are fully investigated, the treatment strategy such as surgical procedure or perioperative chemotherapy will be better utilized. 

## Figures and Tables

**Figure 1 cancers-15-02142-f001:**
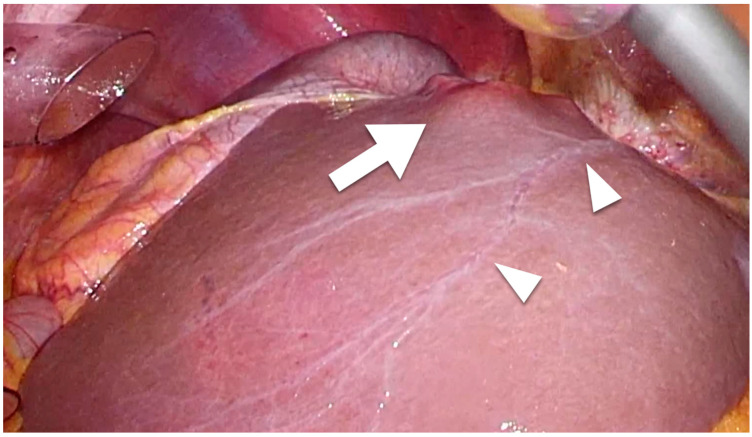
Laparoscopic findings of colorectal liver metastasis. Arrowheads show the capsular (superficial) lymphatic vessels. Tumor (arrow) that has invaded a lymphatic vessel can easily metastasize.

**Figure 2 cancers-15-02142-f002:**
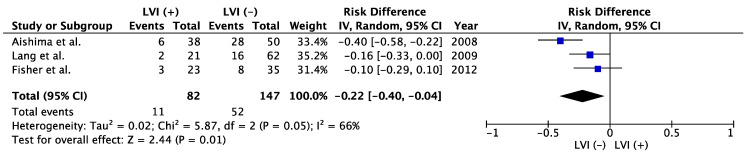
Meta-analysis of studies evaluating the impact of lymphatic vessel invasion on 5-year overall survival for intrahepatic cholangiocarcinoma. CI, confidence interval; IV, inverse variance; LVI, lymph vessel invasion [38,61,74].

**Figure 3 cancers-15-02142-f003:**
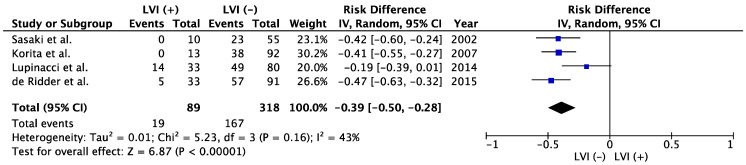
Meta-analysis of studies evaluating the impact of lymphatic vessel invasion on 5-year overall survival for colorectal liver metastasis. CI, confidence interval; IV, inverse variance; LVI, lymph vessel invasion [34,35,36,37].

**Table 2 cancers-15-02142-t002:** Studies evaluating lymphatic vessel invasion in liver cancer.

	Author	Published Year	Period	LVI	Patient Number	Tumor Size, cm	Tumor Number	Poor Histological Type	Vascular Invasion	Lymphadenectomy	LNM	5-Year OS	*p*
ICC	Aishima et al. [61]	2008	1986–2005	positive	38	>4 cm, 21 (23.9%)	NR	28 (31.8%)	NR	NR	27 (30.1%)	16%	<0.0001
				negative	50							55%	
	Lang et al. [74]	2009	1998–2006	positive	21	>5 cm, 70 (84.3%)	multiple, 36 (43.4%)	24 (28.9%)	35 (42.2%)	NR	28 (33.7%)	8%	0.006
				negative	62							26%	
	Cho et al. [73]	2010	2001–2007	positive	22	>5 cm, 36 (57.1%)	multiple, 4 (6.3%)	32 (53.3%)	30 (47.6%)	44 (69.8%)	13 (29.5%)	MST 9	0.008
				negative	41							MST 23	
	Fisher et al. [38]	2012	2000–2010	positive	23	6.5 (1.3–21)	multiple, 12 (21%)	19 (33%)	PNI, 22 (38%)	38 (66%)	13 (22%)	14.3%	0.02
				negative	35							22.2%	
	Lurje et al. [72]	2019	2011–2016	positive	13	≥T2, 17 (28.3%)	multiple, 20 (28.2%)	14 (23.7%)	18 (30.5%)	71 (100%)	24 (40.0%)	MST 4	0.003
				negative	58							MST 40	
CRLM	Sasaki et al. [35]	2002	1982–2000	positive	10	≥5, 18 (26.9%)	≥4, 7 (10.4%)	non-well, 28 (41.8%)	PVI, 15 (23.1%)	NR	primary lesion, 3 (30.0%)	0%	<0.01
				negative	55						primary lesion, 36 (65.5%)	42.3%	
	Korita et al. [37]	2007	1990–2004	positive	13	NR	NR	NR	PVI, 38 (36%)	17 (16.2%)	hepatic node, 3 (23%)	0%	<0.0001
				negative	92						hepatic node, 4 (4%)	41%	
	Lupinacci et al. [36]	2014	2000–2010	positive	33	≥5, 45 (40.0%)	multiple, 65 (58.0%)	NR	49 (43%)	NR	primary lesion, 66 (58.0%)	42%	0.015
				negative	80							61%	
	de Ridder et al. [34]	2015	1992–2011	positive	33	>5, 31 (25%)	all solitary	NR	46 (37.1%)	NR	primary lesion, 70 (56.5%)	14%	0.013
				negative	91							62.2%	

CRLM, colorectal liver metastasis; ICC, intrahepatic cholangiocarcinoma; LNM, lymph node metastasis; LVD, lymphatic vessel density; MST, median survival time; NR, not reported; OS, overall survival; PNI, perineural invasion; PVI, portal vein invasion.

## Data Availability

The data presented in this study are available on request from the corresponding author.

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
