# Peer review of "Prognostic Role of the Intrahepatic Lymphatic System in Liver Cancer"

_cancers, 2023, doi:10.3390/cancers15072142_

Round 1

Reviewer 1 Report

This is a very well written systematic review on an area that is significantly understudied. The review also includes a meta-analysis component and hence needs to address  some specific issues

a. Define exclusion criteria for study selection. LNM has also been defined extensively in hilar/perihilar cholangiocarcinoma. If these were not included please provide justification.

b. What was the literature search strategy and what was the study period?

c. Additional details and references about the statistical methods used should be provided. It is not clear where those were used (Figure 2?) and the conclusions from the statistical analysis of the studies needs to be further elaborated in the discussion section. Figure 2 and its conclusions need to be explained further.

d. References need to be provided for the statistical methods employed

e. Conclusion is too brief and should provide some future perspectives, current limitations and highlight the major significance of this systematic review/meta-analysis.

Author Response

a. Define exclusion criteria for study selection. LNM has also been defined extensively in hilar/perihilar cholangiocarcinoma. If these were not included please provide justification.

Thank you for the comments. We explained the exclusion criteria as follows;

Because the studies investigating the prognostic impact of LNM in liver tumors are enough reported, we excluded the studies only focused on LNM and included the studies that combined assenssment of LVI/lymphangiogenesis. We also excluded the studies about hilar/perihilar cholangiocarcinoma, as most of the main lesions are in extrahepatic bile duct.

b. What was the literature search strategy and what was the study period?

 Thank you for the comments. We added the literature search strategy as follows;

Literature searches of the Cochran database and PubMed were performed for articles published from January 2000, to February 2023. The search strategy “((liver tumor) OR (liver cancer) OR (hepatocellular carcinoma) OR HCC OR (intrahepatic cholangiocarcinoma) OR ICC OR IHCC OR (liver metastasis) OR (metastatic liver tumor)) AND ((Lymphatic vessel invasion) OR (lymph vessel invasion) OR lymphangiogenesis OR (lymph vessel density) OR (lymphatic vessel density) OR podoplanin OR D2-40)” was used and developed by one reviewer (KS). Study selection was performed by two reviewers (KS, KT). Additionally, a manual search of the references of included studies in this review was conducted and the studies which were suitable for this review were also included.

c. Additional details and references about the statistical methods used should be provided. It is not clear where those were used (Figure 2?) and the conclusions from the statistical analysis of the studies needs to be further elaborated in the discussion section. Figure 2 and its conclusions need to be explained further.

Thank you for the comments. We added the detail of meta-analysis of ICC and CRLM as follows;

In Results section of ICC;

Meta-analysis showed that positive LVI in patients with resected ICC was associated with poorer OS than negative LVI (Fig. 2). Five-year OS was 35.3% in LVI (-) group and 13.4% in LVI (+) group (risk difference -0.22; 95% confidence interval -0.40 to -0.04; P = 0.01, Fig. 2).

In discussion section of ICC;

The studies included in meta-analysis showed a little heterogeneity (I2 = 66%), LVI (+) showed significantly poor 5-year OS than that of LVI (-) (P = 0.01, Fig. 2). Therefore, we should focus more on intrahepatic pathological LVI in ICC patients.

In Results section of CRLM;

Among cases with CRLM, LVI represented a poor prognosticator in meta-analysis (Fig. 3). Meta-analysis showed significantly better 5-year OS in LVI (-) group than that of LVI (+) (52.5% vs 21.3%; risk difference -0.39; 95% confidence interval -0.50 to -0.28; P < 0.00001 Fig. 3).

In discussion section of CRLM;

Actually, the present meta-analysis without heterogeneity (I2 = 43%) showed significantly poor 5-year OS in LVI (+) patients than that of LVI (-) (P < 0.00001, Fig. 3). Therefore, further study evaluating LVI/lymphangiogenesis and prognosis of CRLM may develop the tailored surgical procedure such as appropriate hepatic lymph node dissection or precise hepatectomy in the future.

d. References need to be provided for the statistical methods employed.

Thank you for the comments. We added the following references for the statistical methods.

54. McGowan J, Sampson M, Salzwedel DM, Cogo E, Foerster V, Lefebvre C. PRESS Peer Review of Electronic Search Strategies: 2015 Guideline Statement. J Clin Epidemiol. 2016 Jul;75:40-6. doi: 10.1016/j.jclinepi.2016.01.021. Epub 2016 Mar 19. PMID: 27005575.

57. Stewart LA, Clarke MJ. Practical methodology of meta-analyses (overviews) using updated individual patient data. Cochrane Working Group. Stat Med. 1995 Oct 15;14(19):2057-79. doi: 10.1002/sim.4780141902. PMID: 8552887.

58. Higgins JP, Thompson SG. Quantifying heterogeneity in a meta-analysis. Stat Med. 2002 Jun 15;21(11):1539-58. doi: 10.1002/sim.1186. PMID: 12111919.

59. DerSimonian R, Laird N. Meta-analysis in clinical trials. Control Clin Trials. 1986 Sep;7(3):177-88. doi: 10.1016/0197-2456(86)90046-2. PMID: 3802833.

e. Conclusion is too brief and should provide some future perspectives, current limitations and highlight the major significance of this systematic review/meta-analysis.

Thank you for the comments. As the reviewer suggensted, we modified conclusion as follows;

LVI and lymphangiogenesis have prognostic impacts for liver cancer, but the correlations with LNM remain unclear. Further large-scale studies are warranted. In other words, to reveal the association between LVI/lymphangiogenesis and LNM/prognosis, thorough pathological evaluation of LVI/lymphangiogenesis using D2-40 antibody should be performed in liver cancer as well as other gastroenterological cancer. However, the definition of LVI/lymphangiogensis needs further discussion. Nevertheless, if the prognostic role of intrahepatic lymphatic system and mechanism of lymphatic spread of carcinoma are fully investigated, the treatment strategy such as surgical procedure or perioperative chemotherapy will be better utilized.

Reviewer 2 Report

Congratulations for the subject matter and its peculiarities, for the extensive description of the anatomy of the lymphatic vessels of the liver and the dynamics of the lymph, and the consequent pathological correlations, well described. All the small series described in the text and reported in the “huge” bibliography are surgical, and all the series  reported LVI and lymphangiogenesis as poor prognostic factors. My thoughts, as an interventional radiologist, is that many patients affected by liver cancers may be eligible for alternative local therapies (TAE, TACE, TARE, thermal ablation etc.). These patients, even if undergone to percutaneous biopsy, can never be studied so thoroughly, reducing the possibility of determining all risk factors related to their disease. Unfortunately the lack of extensive series underlines the need for future studies on this topic.

Author Response

Thank you  for your valuable comments.
As you pointed out, we believe that further improvement of treatment strategies can be expected by evaluating and reporting more detailed pathological features in liver cancer by our surgeons.

Therefore, we  modified the conclusion as follows;

LVI and lymphangiogenesis have prognostic impacts for liver cancer, but the correlations with LNM remain unclear. Further large-scale studies are warranted. In other words, to reveal the association between LVI/lymphangiogenesis and LNM/prognosis, thorough pathological evaluation of LVI/lymphangiogenesis using D2-40 antibody should be performed in liver cancer as well as other gastroenterological cancer. However, the definition of LVI/lymphangiogensis needs further discussion. Nevertheless, if the prognostic role of intrahepatic lymphatic system and mechanism of lymphatic spread of carcinoma are fully investigated, the treatment strategy such as surgical procedure or perioperative chemotherapy will be better utilized.

Reviewer 3 Report

Perfect, work of interest for more studies

Author Response

Thank you for the comment. 

We are committed to further research on the lymphatic system and liver cancer.